# Change in nutritional status of urban slum children before and after the first COVID-19 wave in Bangladesh: A repeated cross-sectional assessment

**Hayman Win**[1,2]*, **Sohana Shafique**[3], **Nicole Probst-Hensch**[1,2], **Günther Fink**[1,2]

**1** Swiss Tropical and Public Health Institute, Basel, Switzerland, **2** University of Basel, Basel, Switzerland, **3** Health Systems and Population Studies Division, International Centre for Diarrheal Disease Research, Dhaka, Bangladesh

* hayman.win@swisstph.ch

**Data Availability Statement:** The survey data used in this research is exclusively owned by the Asian Development Bank (ADB) and can be made

## Abstract

The onset of COVID-19 severely disrupted economies and increased acute household food insecurity in developing countries. Consequently, a global rise in childhood undernutrition was predicted, especially among vulnerable populations, but primary evidence on actual changes in nutritional status remained scarce. In this paper, we assessed shifts in nutritional status of urban slum children in Bangladesh pre- and post- the country's first wave of COVID-19 and nationwide lockdown. We used two rounds of cross-sectional data collected before and after the pandemic's first year in two large slum settlements (Korail and Tongi) of Dhaka and Gazipur, Bangladesh (n = 1119). Regression models estimated pre-post changes in: 1) predictors of childhood undernutrition (household income, jobs, food security, dietary diversity, healthcare utilization, and hand hygiene); and 2) under-five children's nutritional status (average height-for-age z-score (HAZ) and weight-for-height z-score (WHZ), stunting, and wasting). Subgroup analysis was done by household migration status and slum area. Over the sample period, average monthly household income dropped 23% from BDT 20,740 to BDT 15,960 (β = -4.77; 95% CI:-6.40, -3.15), and currently employed fathers slightly declined from 99% to 95% (β = -0.04; 95% CI:-0.05, -0.02). Average HAZ among the slum children improved 0.13 SD (95% CI: 0.003, 0.26). Among non-migrant children in Tongi, the odds of stunting increased (OR = 2.01, 95% CI: 1.16, 3.48) and average WHZ reduced -0.40 SD (95% CI: -0.74, -0.06). Despite great economic hardship, and differential patterns of representativeness by household geography and migration status, slum children in Bangladesh generally demonstrated resilience to nutritional decline over the first year of the pandemic. While underlying threats to nutritional deterioration persisted, considerable job and income recovery in the post-lockdown period appeared to have cushioned the overall decline. However, as the pandemic continues, monitoring and appropriate actions are needed to avert lasting setbacks to Bangladesh nutritional progress.

available at its discretion for non-commercial academic research purposes. Requests for data can be made to the ADB Project Officer for "TA 9402-REG: Developing Impact Evaluation Methodologies, Approaches, and Capacities in Selected Developing Member Countries (Subproject 3)" through: https://www.adb.org/projects/46185-004/main.

**Funding:** The authors received no specific funding for this work.

**Competing interests:** The authors declare that they have no competing interests.

## Introduction

Following the onset of the Coronavirus Disease 2019 (COVID-19) pandemic in March 2020, many governments worldwide imposed nationwide 'lockdowns' to cope with a rising wave of infections. While the measures saved lives, the lockdowns severely disrupted economic activities and affected the livelihoods of millions of people across socioeconomic strata [1, 2]. In low- and middle-income countries (LMICs), nearly one-third of households suffered employment losses, and more than two-thirds experienced income drops a few months into the crisis [3]. The latest estimates suggest that the pandemic pushed an additional 97 million people—mostly in South Asia—into extreme poverty in 2020 [4], and the number of people suffering acute food insecurity potentially doubled to 265 million in LMICs [5–7]. Consequently, a significant rise in global childhood malnutrition was widely anticipated, reversing past gains and threatening longer-term developmental deficits in children [8–12], along with potentially severe repercussions on child mortality [13, 14].

While COVID-19 invariably affected everyone's lives everywhere [2], the poor in poor countries were disproportionately impacted by the crisis [15, 16]. The overall economic toll was also larger in urban than in rural areas [17, 18]. In Bangladesh, the first full nationwide lockdown was in effect from 26 March to 28 May 2020, following the reporting of its first confirmed COVID-19 case on 8 March 2020. The lockdown upended the lives of urban slum dwellers due to preexisting vulnerabilities [18–20]. Observing stay-at-home orders was challenging for slum dwellers amid realities of overcrowding and poor living conditions [21, 22]. Moreover, livelihoods of slum households were gravely affected due to their large dependence on informal, daily-wage, and other 'COVID-vulnerable' occupations that were disproportionately affected by the economic shutdown [18, 23]. During the lockdown, 71% of slum household main earners faced complete job loss, and 75% of slum households experienced income drops [24]. With limited savings and access to safety nets, nearly 50% of slum households reduced food consumption to cope with income loss [24]. The drastic rise in household food insecurity [16], along with disruptions in food supply chains [25] and routine health and nutrition services [26], during the lockdown implied potentially large declines in the nutritional status of slum children, who already bore the preponderant stunting burden in Bangladesh [27, 28].

While the economic fallout from COVID-19 has been widely expected to increase all forms of child malnutrition in LMICs, including in Bangladesh [29], the primary evidence around longer-term nutritional changes remains limited [17, 30]. Existing research has largely relied on phone surveys, as collection of anthropometric and household data was mostly infeasible amid COVID-19 movement restrictions. In this paper, using newly collected anthropometric and household behavior data, we help respond to the existing gap in evidence on the actual change in the nutritional status of slum children pre- and post- the first COVID-19 wave and nationwide lockdown in Bangladesh.

## Methods

### Study design and setting

This is an observational study based on two rounds of newly collected cross-sectional data. The first survey was implemented from 10 February to 21 March 2020, shortly before the first lockdown period (23 March to 30 May 2020). The second survey took place one year following the initial lockdown during the same season (23 February-27 March 2021). During the first nationwide lockdown, all offices, schools, businesses, and modes of travel were closed down, and the population was advised to stay-at-home unless for meeting essential requirements

[31]. Following the lockdown, risk-based zone-wise restrictions were in place [32], but many economic activities resumed with the government's early implementation of countercyclical measures and stimulus packages targeting key economic sectors [33]. In between the survey periods, there were also institutional and community relief efforts with food, cash, hygiene, and livelihood assistance for affected households, especially in urban areas [25, 34, 35].

The study setting covered two large slum areas: Korail and Tongi (**S1 Fig**). Korail is located in Dhaka North City Corporation and is one of the largest slums in Bangladesh. Most of Korail residents were traditionally migrants from some of the poorest parts of Bangladesh, living below the poverty line and engaged in some of the lowest-paid jobs [36]. Tongi is located in Gazipur City Corporation, adjacent to capital Dhaka. A large proportion of Tongi slum populations were also migrants from other parts of the country who have moved there for work. Tongi traditionally had low rates of unemployment, with men mostly working in service or industrial sectors, and a large share of women working in the ready-made garment industry [37].

### Data collection and ethical approval

The surveys were originally designed and implemented as a baseline evaluation—funded by the Asian Development Bank under *TA-9402 REG*: *Developing Impact Evaluation Methodologies*, *Approaches*, *and Capacities in Selected Developing Member Countries*—for assessing the impact of nutrition interventions under the government *Urban Primary Health Care Service Delivery Project (UPHCSDP)*. The UPHCSDP baseline survey was nested in the Urban Health and Demographic Surveillance System (UHDSS), which covered approximately 30,000 households in five large slum areas in three city corporations of Dhaka Division [38]. The UPHCSDP survey had a target sample size of 2,100 children, with two-stage stratified random sampling. The sample size was calculated based on detecting a minimum average length/height-for-age z-score (HAZ) difference of 0.25 standard deviations (SDs) between project intervention and control groups, with a 5% significance level, 80% power, and design effect of 3.05. All 54 UHDSS slum clusters were selected in the first stage, and in the second stage, 39 under-five children (0–59 months) were randomly selected from each cluster based on updated household list of the most-recent UHDSS surveillance round.

Against the rapidly evolving backdrop of COVID-19, two repeated attempts were made to complete the UPHCSDP baseline data collection: The first survey began early February 2020—well before reporting of the first COVID-19 case in Bangladesh and movement restrictions—and collected 346 observations before its suspension due to COVID-19's spread to the country and ensuing nationwide lockdown. The second survey began after one year, following careful monitoring of the local epidemiological situation and obtaining necessary institutional clearances from research, government, and funding agencies. The second round followed the same sampling approach but used an updated UHDSS listing, and newly selected the households with under-five children for interview. The repeat-survey collected 773 observations, prior to coinciding with the onset of a second COVID-19 wave in Bangladesh, and the fieldwork was suspended following the government announcement of a second nationwide lockdown. (**S1 Table** shows distribution of sampled households by survey round).

The data was collected by the International Centre for Diarrheal Diseases Research, Bangladesh (icddr,b), a local research agency. Tab-based, pre-coded, and pre-tested structured questionnaires were used to collect information on a range of household and individual characteristics. The questions were adapted from publically available sources, including the Bangladesh Demographic and Health Survey, Bangladesh Urban Health Survey, Food Security and Nutrition Surveillance Project (Bangladesh), and the Household Food Insecurity and

Access Scale (HFIAS) that was validated for Bangladesh. Local field enumerators conducted interviews with mothers or caretakers in their homes in the local language. Anthropometric measurements were taken by field interviewers and supervisors trained by a nutritionist from icddr,b.

The surveys—including an addendum to the research protocol with additional pandemic-related research questions in the second round—were approved by the Ethical Review Committee and the Research Review Committee of the Institutional Review Board of icddr,b on ethical issues and technical competence. Written informed consent was obtained from each respondent prior to the interview. Before taking consent, interviewers explained to participants all relevant information on the consent form, including research purpose, assurance of confidentiality, and their right to withdraw at any time without further obligation. In cases where respondents could neither read nor write, interviewers read aloud the consent form in entirety and thumbprints were taken. Additional information regarding the ethical, cultural, and scientific considerations specific to inclusivity in global research is included in the **S1 Checklist**.

Over the course of the pandemic, implementation of field activities followed icddr,b's strict policies and procedures for managing project related COVID-19 risks, as well as prevailing national laws and public health guidelines. The Resumption of Projects Committee (RPC) at icddr,b internally approved the fieldwork recommencement based on considerations of local epidemiological situation, field safety protocols, requisite staff trainings, and availability of personal protective equipment (PPE). Permission to resume survey activities was also granted by the UPHCSDP project office at the Ministry of Local Government. Prior to data collection, all field staff were trained in appropriate biosafety and community engagement measures in accordance with icddr,b's standard operating procedures. Relevant PPEs (including face masks, face shields, gloves, and sanitizers) were provided for all fieldworkers and research participants and replenished as needed. During the survey, both interviewers and respondents were required to wear appropriate levels of PPEs, and all reusable anthropometry equipment were cleaned and disinfected before and after each measurement to protect the research participants. A dedicated field staff was assigned to monitor and ensure compliance with COVID-19 safety protocols and to report any incidents. The RPC also monitored the community case positivity rate, statuses of PPE and field staff, and project compliance with safety protocols on a weekly basis.

### Variables

The main outcome measures were stunting and wasting as binary variables. Stunting was defined as HAZ of more than two SDs, and wasting was defined as weight-for-height z-score (WHZ) of more than two SDs, below the 2006 WHO Child Growth Standards median. In addition, we showed average HAZ and WHZ as continuous variables to capture absolute shifts in the local population. We used the Stata software built-in package *zscore06* to transform child height and weight measurements into z-scores. All height and weight measurements were completed using equipment (SECA weighing scales and portable height boards) on lent from the UNICEF Bangladesh office. We excluded from our analysis 23 (2.1%) observations with extreme HAZ values (<-6 and >6 SDs), and 23 (2.1%) observations with extreme WHZ values (<-5 and >5 SDs), following the WHO-recommended acceptable range [39, 40].

As intermediary outcomes, we considered the underlying determinants of childhood undernutrition presented in the perennial UNICEF framework [41], which were likely directly or indirectly affected by the COVID-19 crisis. We included factors of household assets, household income, parental job status, household food security, child's dietary diversity, healthcare

seeking, and hand hygiene. Household asset scores—reflecting household ownership of durable assets and amenities—were derived using a principal components analysis procedure in the pooled data (both survey rounds), and divided into quintiles. The monthly household income was respondent-reported approximate amount in the local currency, Bangladeshi Taka or BDT (US$1 = 85 BDT). On job status, we showed both mother's and father's current employment in binary terms and by main categories of occupation; mothers and fathers were considered 'currently working' if reportedly being engaged in any kind of income-generating work, whether wage- or self-employment, at the time of the interview.

We assessed household food security using the HFIAS, with continuous scores ranging from 0–27, and higher score indicating higher food insecurity [42]. From the HFIAS, we also constructed categorical prevalence indicators, placing households into four levels of food insecurity: food secure, mildly food insecure, moderately food insecure, and severely food insecure [42]. To assess the child's dietary intake, we considered minimum dietary diversity and protein intake based on caregiver's 24-hour recall. Children (older than 6 months) were considered to have met minimum dietary diversity if they received foods from at least four out of the seven food groups (excluding breastmilk) recommended in infant and young child feeding (IYCF) [43]. Child's protein intake was a continuous variable indicating the number of animal-source foods consumed the previous day [44]. Healthcare utilization was measured as binary indicators based on caregivers' report of 1) whether any medical advice was sought when the child was last sick with fever or cough, 2) whether antenatal care (ANC) was received for the child while in-utero, and 3) whether the child was born at home or at a health facility. Categorization of facility-based births included birth in public, private, or non-government organization (NGO)-run health facilities, as well as community-based 'birthing huts' run by BRAC NGO. Assessment of hand hygiene was presented as binary variables based on interviewers' observation of 1) whether a handwashing place was in the dwelling area, and 2) whether soap or detergent was present at the handwashing place.

Covariates adjusted for were slum area, migration status, and parental education. A household was considered as 'migrant' in the slum area if it had moved from another place, regardless of length of stay. Maternal and paternal education were highest class completed in years, and coded in binary terms of 'no or primary level education' (0–5 years) and 'secondary or higher-level education (6 or more years).

## Statistical analysis

First, we provide descriptive statistics on household and individual background characteristics by survey year. We used Pearson chi-square (for binary and categorical data) and Adjusted Wald (for continuous data) tests to assess the statistical significance (*p* values) of differences between baseline (prior to first lockdown in 2020) and endline (one year after lockdown in 2021). Second, ordinary least squares (OLS) regression models estimated the overall changes (with 95% CI) in intermediary outcomes (for multi-categorical variables, we created binary indicators for each category and showed the level-specific changes). Third, logistic (for stunting and wasting) and OLS (for HAZ and WHZ) regression models estimated the changes in child anthropometric outcomes. While we showed unadjusted estimates as our main result depicting average changes in child anthropometric outcomes, we also included estimates controlling for parental education–-a partial measure of socio-economic status (SES)—to assess how results varied with SES-related changes in population composition. Next, given some disparities in proportion of migrant households and respondents by slum area between baseline and endline, we conducted subgroup analyses of changes in anthropometric and intermediary outcomes by migrant status and slum area. Kernel density plots graphically compared

**Table 1. Background characteristics of respondents at baseline and endline.**

| Background characteristics | | Baseline-2020 (*n* = 346) | Endline-2021 (*n* = 773) | *P*-value* |
|---|---|---|---|---|
| **Household** | | | | |
| Slum location (%) | Korail | 48.27 | 44.63 | 0.26 |
| | Tongi | 51.73 | 55.37 | |
| Migration status (%) | Migrant | 66.18 | 40.88 | <0.001 |
| Religion of household head (%) | Islam | 99.71 | 98.71 | 0.12 |
| | Hindu | 0.29 | 1.29 | |
| Household size (Mean ±SD) | | 4.89 ±1.96 | 4.86 ±1.69 | 0.79 |
| Drinking water source (improved) (%) | | 100.00 | 100.00 | - |
| Water not available at source for at least 1 full day (last 2 weeks) (%) | | 24.57 | 18.10 | 0.013 |
| Toilet type (improved) (%) | | 99.71 | 93.47 | <0.001 |
| Toilet type (shared with other households) (%) | | 66.38 | 69.36 | 0.32 |
| Main source of cooking fuel (%) | Solid fuels | 15.41 | 19.79 | 0.08 |
| Distance of nearest public/NGO health facility (%) | Within 1 km | 87.24 | 87.45 | 0.93 |
| **Child** | | | | |
| Sex (%) | Female | 47.40 | 48.38 | 0.76 |
| Age (in months) (Mean ±SD) | | 28.84 ±16.30 | 29.58 ±16.10 | 0.48 |
| Birth order (Mean ±SD) | | 1.90 ±0.95 | 1.97 ±1.07 | 0.28 |
| **Mother** | | | | |
| Age (years) (Mean ±SD)[a] | | 26.61 ±5.79 | 27.00 ±5.82 | 0.30 |
| Education completed (%)[a] | Secondary/ higher level | 42.23 | 42.80 | 0.86 |
| Literacy (%)[b] | Cannot read at all | 21.11 | 20.25 | 0.45 |
| Media exposure (at least once a week to newspaper/radio/TV) (%)[b] | | 85.63 | 86.40 | 0.74 |
| Age at cohabitation with (first) husband (Mean ±SD)[b] | | 16.19 ±2.09 | 16.48 ±2.83 | 0.06 |
| Number of children ever born (Mean ±SD)[b] | | 2.00 ±0.96 | 2.09 ±1.14 | 0.17 |
| **Father (mother's husband)** | | | | |
| Age (years) (Mean ±SD)[b] | | 32.93 ±6.89 | 33.11 ±6.63 | 0.70 |
| Education completed (%)[b] | Secondary/ higher level | 49.56 | 44.35 | 0.12 |

*Denotes statistical significance of difference in mean levels between baseline and endline.

[a] includes mother's background information reported by non-mother caretaker respondents

[b] restricted to mother-only respondents.

distributions of continuous anthropometric outcomes by subgroup. We assumed statistical significance at the 5% threshold ($p \leq 0.05$). We used *Stata 15.1* (Stata Corps) for data analyses and applied clustered standard errors (using the *cluster* command in Stata) in all regression models to adjust for sample clustering and address heteroscedasticity in variance. Analyses excluded missing data.

## Results

A total of 1,119 child-caregiver dyads were assessed. **Table 1** shows the background characteristics of the respondents by survey year. Overall, respondent characteristics appeared similar in baseline and endline. The proportion of respondents from Tongi increased slightly in the endline survey (see **S1 Table** on additional sampling clusters), and the share of migrant households decreased from 66% to 41% ($p < 0.001$). Water interruptions at source reduced from 25% to 18% ($p = 0.013$), while access to improved toilet type marginally declined from nearly 100% to 94% ($p < 0.001$).

**S2 Fig** shows the monthly in- and out-migration patterns during February 2020-March 2021 for Korail and Tongi slum areas based on UHDSS data. It shows that in-migration declined sharply for both slums during the lockdown period (March-May 2020) and increased again beginning June 2020 after the restrictions were lifted. At the beginning of 2021, Tongi experienced a large spike in out-migration, while in-migration surged in Korail.

**Table 2** presents changes in predictors of childhood undernutrition between baseline and endline. One year after the pandemic's onset, the average monthly household income dropped by 23% from BDT 20,740 to BDT 15,960 ($\beta$ = -4.77, 95% CI: -6.40, -3.15). The overall proportion of currently employed fathers decreased only 4% points ($\beta$ = -0.04; 95% CI: -0.05, -0.02), but occupational changes were apparent. Employment in service sector jobs decreased, while daily-wage work, such as day laborers and rickshaw pullers, increased. The proportion of 'severely food insecure' households increased 10% points ($\beta$ = 0.10; 95% CI: 0.04, 0.17). There were also drops in average child dietary diversity, with the proportion consuming adequate number of food groups declining 26% points ($\beta$ = -0.26, 95% CI: -0.37, -0.16), and daily intake of animal-based proteins dropping roughly from 2 to 1 sources ($\beta$ = -0.61, 95% CI: -0.75, -0.48). However, the share of households with a handwashing place observed increased to nearly universal at the endline. In terms of health seeking, there was 7% point drop ($\beta$ = -0.07, 95% CI: -0.11, -0.03) in children that received ANC, and 9% point rise ($\beta$ = 0.09, 95% CI: 0.02, 0.15) in home births. On the other hand, there was 14% point increase ($\beta$ = 0.14, 95% CI: 0.11, 0.18) in children seeking medical advice when they last had fever or cough symptoms. However, this increase in health seeking was alongside a rise in informal private care and a decrease in care from formal facilities as access points (see **S3 Fig**).

**Table 3** shows estimated changes in child anthropometric status between baseline and endline. Average HAZ increased by 0.13 SD (95% CI: 0.003, 0.26) over the sample period, although the odds of stunting did not significantly change (OR = 0.81, 95% CI: 0.62, 1.05); conditioned on parental education, the estimated increase in HAZ was 0.17 SD (95% CI: 0.08, 0.27). When controlled for parental education, average WHZ decreased -0.16 SD (95% CI: -0.30, -0.01), and the odds of wasting were 42% higher (OR = 1.42, 95% CI: 1.05, 1.93) for children at endline compared to baseline.

**Table 4** shows changes in child nutritional status by subgroups according to household slum area and migration status. Between baseline and endline, there was deterioration in average WHZ only among Tongi non-migrant children ($\beta$ = -0.40 SD, 95% CI: -0.74,-0.06), although the change in their odds of wasting was not significant. Among Korail migrant children, the odds of wasting increased, but only when adjusted for parental education (OR = 1.90, 95% CI: 1.07, 3.37). The odds of stunting also doubled (OR = 2.01, 95% CI: 1.16, 3.48) among Tongi non-migrant children, partly explained by change in parental education (OR = 1.67, 95% CI: 1.02, 2.74); however, the change in average HAZ was insignificant in this group, and the relevant kernel density estimates showed a subtle shift in HAZ distributions (see **S4 Fig**, panel C). Among Korail migrant children, the odds of stunting decreased on average in both unadjusted (OR = 0.62, 95% CI: 0.42, 0.90) and adjusted (OR = 0.62, 95% 0.42, 0.92) models. The improvement in average HAZ was significant only among Korail non-migrant children ($\beta$ = 0.80 SD, 95% CI: 0.39, 1.21), which sample size was notably small at baseline (**S4 Fig**, panel E).

**Fig 1** shows estimated changes in the intermediary variables by subgroup. The drops in mean household wealth quintile and monthly income were larger in Tongi than in Korail (panels A and B). The Tongi non-migrant group had the largest income drop, from having the highest level at baseline to the lowest level at the endline (panel B). The overall reduction in the share of currently working fathers between baseline and endline was small and similar across

**Table 2. Estimated change in underlying predictors of childhood undernutrition between baseline and endline.**

| Predictor variables | Baseline (2020)* | Endline (2021)* | P-value** | Change coefficient (95% CI)*** | N§ |
|---|---|---|---|---|---|
| Household asset & income | | | | | |
| Household assets (mean quintile) | 3.12 ±1.34 | 2.94 ±1.45 | 0.038 | -0.185 (-0.574, 0.204) | 1119 |
| Household monthly income (mean, in BDT '000s) | 20.74 ±10.21 | 15.96 ±8.66 | <0.001 | -4.773 (-6.400, -3.147) | 1046 |
| Employment | | | | | |
| Father's work status% | | | | | 1038 |
| Currently working | 99.12 | 95.40 | 0.002 | -0.037 (-0.053, -0.022) | |
| Father's work type (if working)% | | | | | 1003 |
| Day laborer | 12.68 | 16.87 | 0.08 | 0.042 (0.000, 0.084) | |
| Self-employed/ own business | 29.50 | 26.96 | 0.40 | -0.025(-0.089, 0.038) | |
| Factory/ garment worker | 6.19 | 6.63 | 0.79 | 0.004(-0.044, 0.052) | |
| Rickshaw puller/ vehicle driver | 12.09 | 18.52 | 0.009 | 0.064(0.005, 0.123) | |
| Service | 38.64 | 29.37 | 0.003 | -0.093(-0.171, -0.015) | |
| Other | 0.59 | 1.51 | 0.21 | 0.009(-0.002, 0.020) | |
| Mother's work status% | | | | | 1050 |
| Currently working | 15.70 | 15.01 | 0.77 | -0.007 (-0.052, 0.038) | |
| Mother's work type (if working)% | | | | | 160 |
| Day laborer | 12.96 | 4.72 | 0.06 | -0.082(-0.221, 0.056) | |
| Domestic worker | 24.07 | 20.75 | 0.63 | -0.033(-0.251, 0.185) | |
| Self-employed/ own business | 24.07 | 34.91 | 0.16 | 0.108(-0.052, 0.269) | |
| Factory/ garment worker | 22.22 | 16.98 | 0.42 | -0.052(-0.178, 0.074) | |
| Service | 11.11 | 18.87 | 0.21 | 0.078(-0.041, 0.196) | |
| Other | 1.85 | 3.77 | 0.51 | 0.019(-0.034, 0.072) | |
| Household food security & child's dietary intake | | | | | |
| HFIAS (0–27, mean score) | 4.29 ±4.71 | 6.05 ±5.83 | <0.001 | 1.765 (0.819, 2.711) | 1116 |
| HFIA status% Food secure | 40.58 | 33.85 | 0.030 | -0.067(-0.153, 0.018) | 1116 |
| Mildly food insecure | 14.78 | 6.36 | <0.001 | -0.084(-0.128, -0.041) | |
| Moderately food insecure | 31.30 | 36.19 | 0.11 | 0.049(-0.048, 0.145) | |
| Severely food insecure | 13.33 | 23.61 | <0.001 | 0.103(0.035, 0.170) | |
| Child ate ≥4 IYCF food groups yesterday% [a] | 67.27 | 41.12 | <0.001 | -0.262 (-0.365, -0.158) | 1045 |
| Animal-source proteins (dairy poducts, flesh foods, eggs), mean # sources ate yesterday[a] | 1.89 ±0.69 | 1.28 ±0.88 | <0.001 | -0.611 (-0.747, -0.475) | 1045 |
| Healthcare seeking | | | | | |
| Sought medical advice when child last sick (fever/cough) % | 78.22 | 92.64 | <0.001 | 0.144 (0.107, 0.182) | 1033 |
| Child received ANC% | 93.62 | 86.81 | 0.001 | -0.068 (-0.111, -0.025) | 1088 |
| Child's place of birth% | | | | | 1112 |
| At home | 40.99 | 49.48 | 0.009 | 0.085 (0.020, 0.150) | |
| Hand hygiene | | | | | |
| Handwashing place observed in dwelling/compound % | 80.64 | 98.82 | <0.001 | 0.182 (0.048, 0.316) | 1111 |
| Presence of soap/detergent at handwashing place% [b] | 68.85 | 71.30 | 0.45 | 0.025 (-0.104, 0.153) | 1016 |

*Values presented in percentages or means ± SDs

**Denotes statistical significance of difference in mean levels between baseline and endline

***regression estimate of change between baseline and endline with clustered standard errors; §Missing and not-applicable values excluded

[a] sample restricted to children older than 6 months

[b] sample restricted to where handwashing place was observed

HFIAS = Household food insecurity and access score; IYCF = infant and young child feeding; ANC = antenatal care.

**Table 3. Estimated changes in child nutritional status between baseline and endline (full sample).**

| N = 1096 § | Mean HAZ | Mean WHZ | Stunted | Wasted |
|---|---|---|---|---|
| Baseline | -1.37 ±1.26 | -0.49 ±1.11 | 29.28% | 7.67% |
| Endline | -1.24 ±1.29 | -0.61 ±1.12 | 25.03% | 9.38% |
| *Estimated change* | OLS (95% CI) | | OR (95% CI) | |
| Unadjusted model | 0.131 (0.003, 0.258)* | -0.124 (-0.275, 0.028) | 0.81 (0.62,1.05) | 1.25 (0.92,1.69) |
| Model adjusted for parental education | 0.173 (0.076, 0.271)* | -0.157 (-0.300,-0.014)* | 0.77 (0.59,1.01) | 1.42 (1.05,1.93)* |

*p ≤0.05; standard errors clustered

§ excludes observations with extreme of biologically implausible HAZ/WHZ z-scores; OLS = ordinary least squares; OR = odds ratio.

all groups (panel C). However, the proportion of currently working mothers increased in Korail, while dropping in Tongi (panel D). The proportion of moderately and severely food insecure households increased across all groups, especially among Tongi migrants (panel E). The proportion of children consuming adequate dietary diversity (at least four food groups) was reduced considerably more in Tongi than in Korail (panel F). Home births increased for all groups at endline compared to baseline, with the largest increase among Tongi migrants (panel G). The increase in the share of households with an observed handwashing place was higher in Korail, which started with a lower baseline, than in Tongi (panel H).

**Fig 2** reports additional information collected at the endline on changes in household food access and hand hygiene due to COVID-19 by slum area. Approximately 92% of households in Tongi reported a change in their household food access compared to 55% in Korail (panel A). A considerably higher proportion of households in Korail reported improved access to hand sanitizers than in Tongi (panel B).

**Table 4. Estimated changes in child nutritional status between baseline and endline (by subgroups according to household migration status and slum area).**

| N § | 155 | 402 | 347 | 192 | 159 | 404 | 345 | 188 |
|---|---|---|---|---|---|---|---|---|
| Group | Not-Migrant, Korail | Not-Migrant, Tongi | Migrant, Korail | Migrant, Tongi | Not-Migrant, Korail | Not-Migrant, Tongi | Migrant, Korail | Migrant, Tongi |
| | **Mean HAZ** | | | | **Mean WHZ** | | | |
| Baseline | -2.28 | -0.95 | -1.50 | -1.39 | -0.59 | -0.20 | -0.62 | -0.59 |
| Endline | -1.48 | -1.06 | -1.32 | -1.28 | -0.65 | -0.60 | -0.64 | -0.55 |
| *Estimated change* | OLS (95%CI) | | | | OLS (95%CI) | | | |
| Crude model | 0.80 (0.39, 1.21)* | -0.12 (-0.47, 0.23) | 0.18 (-0.08, 0.44) | 0.11 (-0.23, 0.45) | -0.06 (-0.43, 0.31) | -0.40 (-0.74,-0.06)* | 0.03 (-0.30, 0.25) | 0.03 (-0.33, 0.39) |
| Model adjusted for parental education | 0.84 (0.37, 1.32)* | 0.02 (-0.25, 0.30) | 0.19 (-0.07, 0.45) | 0.10 (-0.31, 0.52) | -0.01 (-0.32, 0.29) | -0.39 (-0.72, -0.06)* | -0.11 (-0.38, 0.15) | 0.01 (-0.32, 0.34) |
| | **Stunted** | | | | **Wasted** | | | |
| Baseline | 47.62% | 12.63% | 35.62% | 32.53% | 9.52% | 5.26% | 8.33% | 8.86% |
| Endline | 30.60% | 22.48% | 25.37% | 24.77% | 11.59% | 7.12% | 10.95% | 10.09% |
| *Estimated change* | OR (95%CI) | | | | OR (95%CI) | | | |
| Crude model | 0.49 (0.23, 1.00)* | 2.01 (1.16, 3.48)* | 0.62 (0.42, 0.90)* | 0.68 (0.41,1.14) | 1.25 (0.48,3.23) | 1.38 (0.64,3.00) | 1.35 (0.73, 2.52) | 1.15 (0.49, 2.74) |
| Model adjusted for parental education | 0.44 (0.19, 1.03) | 1.67 (1.02, 2.74)* | 0.62 (0.42, 0.92)* | 0.69 (0.37,1.30) | 1.20 (0.45,3.18) | 1.29 (0.55,3.04) | 1.90 (1.07, 3.37)* | 1.11 (0.47, 2.62) |

*p ≤0.05; standard errors clustered

§ excludes observations with extreme of biologically implausible HAZ/WHZ z-scores; OLS = ordinary least squares; OR = odds ratio.

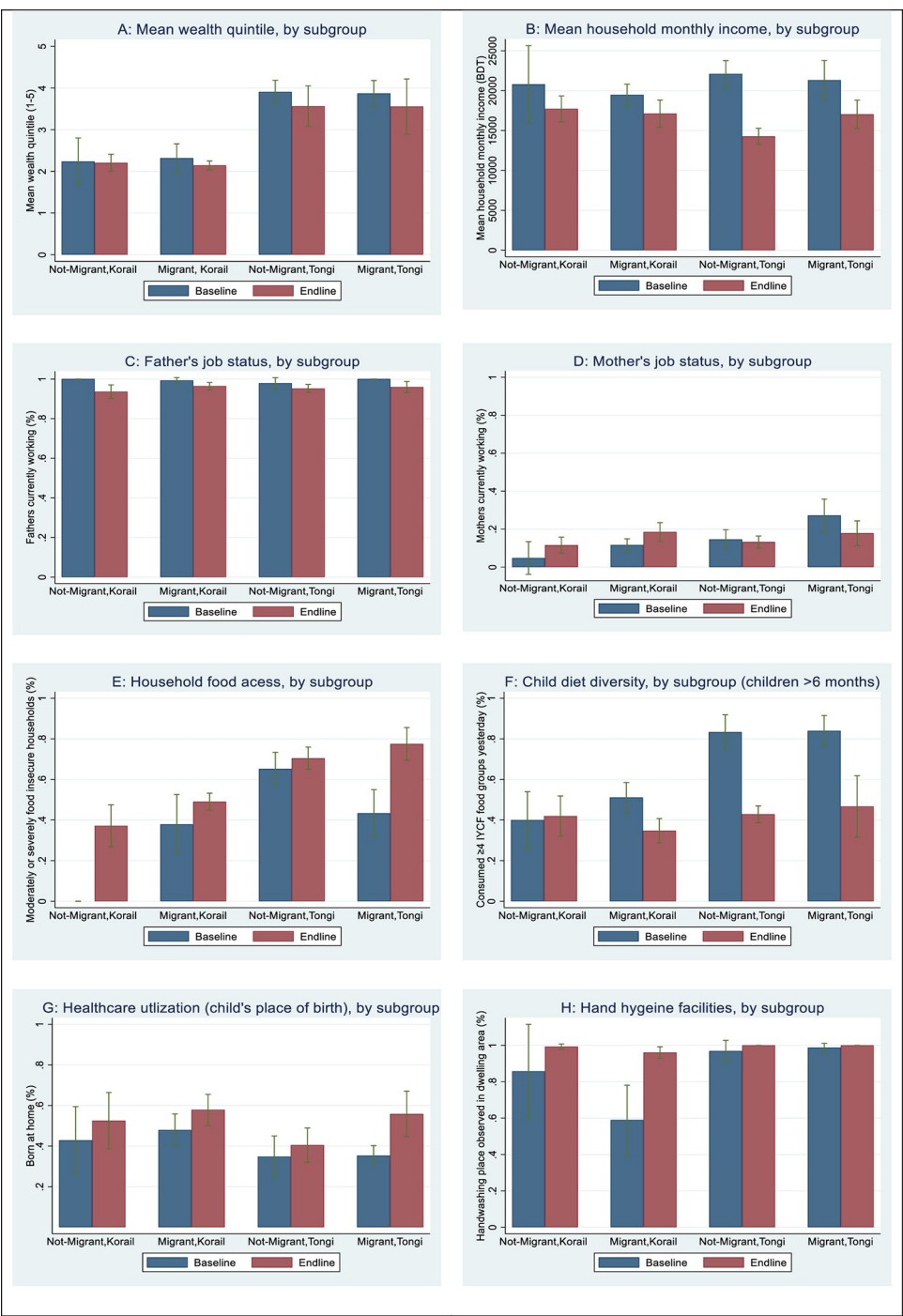

**Fig 1. Estimated changes in intermediary variables between baseline and endline, by subgroup according to slum area and migration status.**

## Discussion

In this paper, we used two rounds of newly collected data to assess changes in nutritional status of Bangladeshi slum children pre- and post- the country's first COVID-19 wave. We found lingering moderate declines in underlying predictors of childhood undernutrition—household

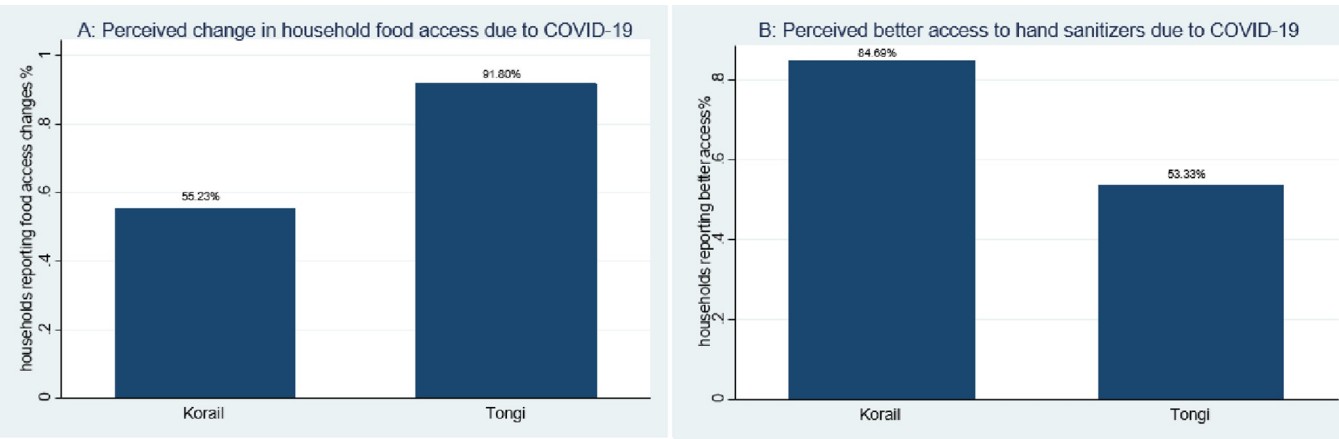

**Fig 2. Perceived changes in household food security and hand hygiene due to COVID-19, by slum area.**

income, food security, child's dietary diversity, and use of maternal health services—one year into the COVID-19 pandemic. However, the overall children's nutritional status largely did not decline, with a marginal improvement in average HAZ, and marginal deteriorations in average WHZ and wasting when adjusted for changes in population. This finding attached two important considerations: First, there were changes in composition of slum populations triggered by COVID-19 related economic migration. Following the economic shock, many urban households moved to cheaper locations, including to other cities or to villages, or in search of alternative work opportunities [45]. When controlled for differences in SES (parental education), our data showed a decline in average WHZ, along with increased odds of wasting, suggesting an overall rise in acute undernutrition among comparable populations. Wasting can be seasonal, or coincide with periods of heightened food security [46]. Second, we found differential patterns of representativeness and variations in child nutritional changes by slum area and household migration status. Among non-migrant children in Tongi, the average WHZ declined considerably, whereas similar degree of WHZ deterioration was not seen elsewhere. The odds of stunting also increased among Tongi non-migrant children (although the change in average HAZ was subtle), while the odds of stunting decreased among children in Korail.

To our knowledge, this study is one of the first to provide primary evidence on COVID-19 related changes in child nutritional status, albeit in a limited setting. As such, there is currently a dearth of literature for comparing our main findings in similar contexts. In general, our findings suggest that the predictions of large increases in childhood undernutrition due to COVID-19 had largely not materialized in this slum population one year into the crisis. Several factors may explain this outcome: First, we may have underestimated the overall adverse effect on nutritional status of slum children due to follow-up loss and 'reverse migration' (i.e. urban migrants returning to rural areas). Research by Power and Participation Research Center (PPRC) estimated this population to be about 10% of slum households, with a vast majority of them (61%) coming from the pre-pandemic 'poor' category [47], and thus presumably with higher food insecurity and nutritional deficits. Second, the role of social and institutional assistance—albeit uneven and with myriad flaws in targeting—likely helped cushion the short-term effect on food security at its most severe period. During the first nationwide lockdown, approximately 61% of urban slum households reported receiving some kind of relief support, and much of the support was in the form of food assistance [48]. The government also initiated measures such as 'open market sale' of foods at subsidized prices [25]. Third, there was swift

resumption of economic activities immediately following the lockdown period, and implementation of stimulus packages that facilitated high level of employment recovery, with many being able to resume their pre-pandemic occupations [48]. Next, although there were considerable occupational shifts, post-lockdown employment recovery seemed high especially among migrants—who comprise a large share of slum populations—which likely mitigated the overall income loss [45]. Nearly 30% of slum dwellers moved at least once in 2020, largely driven by unemployment, and many sought alternative work elsewhere in areas with higher productivity [45, 49].

The differences we found in child nutritional changes by slum area and household migration status may be largely due to varying levels of area-specific economic recovery. For example, UHDSS migration data showed a large surge in out-migration from Tongi at the beginning of 2021, suggesting low economic recovery of the area (see **S2 Fig**). While many migrants moved or relocated to cope with job and income losses, original residents (non-migrants), such as in Tongi, were presumably less willing to move elsewhere, perhaps due to ties with local assets or extended families. Furthermore, our data suggested an increased share of working mothers in Korail, while the same declined in Tongi. The additional household income from mother's work in Korail—and its potential use towards enhancing food security and nutritional benefits for children—could have also contributed to tempering adverse child nutritional outcomes in the slum area.

Our findings of an overall moderate drop in household income and high job recovery among fathers (main earners) were similar to results from representative phone surveys conducted by PPRC in urban slums. The surveys reported rapid economic recovery immediately after easing of lockdown restrictions, and a remaining drop in average household income of 14% and sizeable occupational shifts, mostly into lower skilled occupations, one year into the pandemic [45]. Consistent with our finding on largest income drop among Tongi non-migrants—who were socio-economically better off at baseline—the report of PPRC also commented on larger severity of income drop among the non-poor [45]. This was likely due to variations in type of pre-crisis occupations [18], with Tongi presumably having a larger composition of workers in harder-hit sectors, and therefore having shifted to lower-paid work.

Previous studies found that COVID-19 effects on income and jobs were predictors of food insecurity and diet diversity [50, 51]. Household food insecurity in turn predicted children's linear growth in slum settings [52, 53] and is associated with stunting and wasting in Bangladesh [54]. Along with the drop in household income, our data found continued declines in household food security and child's dietary diversity, with the effects more pronounced in Tongi. PPRC's survey similarly reported 17% lower food expenditure in slums one year into COVID-19, and a majority of slum households forgoing proteins in their weekly diets [45]. Our finding on reduced child dietary diversity also supports COVID-19 impact predictions for other countries [55, 56]. Lower dietary diversity is generally associated with a rise in food insecurity [54], but also as households may choose to consume more staples in lieu of fresh or faster-perishable food items in the pandemic's context [17, 48].

Current evidence—albeit less tenable—also supports the association between healthcare utilization and childhood undernutrition [57–59]. Our study found an increase in homebirths and a decrease in share of children who received ANC, compared to the pre-pandemic situation. A study for urban Dhaka found a significant decrease in health facility visits by pregnant women between February and September 2020, with COVID-19 adversely affecting service utilization despite service provision adaptation [26]. Fears among pregnant women to leave the house and contract COVID-19 at health centers affected service utilization; on the side of health providers, delivering services such as ANC—typically done through outreach—was hampered by increased workload, fear of house visits, and lack of transport [26]. On the other

hand, we found improved water security and handwashing facilities as positive offshoots of the pandemic, although the evidence around contributions of water, sanitation, and hygiene (WASH) elements to child undernutrition remain mixed [60–62].

## Limitations

Our study has several limitations: First, the study uses data from repeated cross-sectional surveys and is therefore is subject to confounding concerns in the change estimations. Use of longitudinal data would better control for unobserved variables or residuals and reduce potential biases in the regression estimates. Second, due to the study's limited geographic focus, representativeness and generalizability of these results to all Bangladesh slums may be limited. Availability of data representative of slum populations, and also at the levels of different slum types and locations, would produce more accurate and generalizable results. Third, while the outcome measures were objective, household food insecurity and questions related to child's dietary intake were based on self-report, and thus subject to reporting and recall biases. Data collection through direct observation or requesting participants to keep food diaries would help reduce some of these errors. Lastly, as the pandemic drags on, the limited duration of the study, with only two (pre- and post-) time points over one year, was insufficient to draw conclusions on continued recovery and resilience of the population to subsequent shocks following the first year [63]. Further studies with longer time-periods, and using panel data with multiple follow-up points per individual, would more accurately assess the nutritional resilience of slum children in the COVID-19 context.

## Conclusion

Despite the great economic hardship ushered by the COVID-19 pandemic and the ensuing first nationwide lockdown, slum children in Bangladesh—who disproportionately bear the country's stunting burden—generally demonstrated resilience to overall nutritional decline over the first year of the pandemic. While the threats to nutritional deterioration (e.g. reduced household income and food security) lingered after one year, there was considerable recovery during the post-lockdown period, compared to drastic drops experienced during the lockdown, which appeared to have cushioned the overall decline. However, as the pandemic unfolds and new restrictions imposed to control future waves, further declines in household income, food security, and use of essential maternal health services could yet translate into permanent adverse consequences for the vulnerable slum children. As such, social protection and safety net programs for slum populations should expand and continue beyond lockdown periods. Social and institutional assistance should also include addressing rise in nutrition vulnerabilities among those previously better off but facing large income losses, and ensure expanding reach to peripheral areas outside Dhaka. Strategies to maintain utilization of essential maternal and child health services during the pandemic would also be important [64].

## Supporting information

**S1 Checklist. Inclusivity in global research.**
(DOCX)

**S1 Table. Sampling distribution comparing baseline and endline.**
(DOCX)

**S1 Fig. Tongi (Gazipur) and Korail (Dhaka North) slum locations.**
(TIF)

**S2 Fig. Total migration events during Jan 2020-March 2021, by slum area.**
(TIF)

**S3 Fig. Place where medical advice was sought when child was last sick with cough or fever.**
(TIF)

**S4 Fig. Kernel density estimates of HAZ and WHZ distributions (full sample and by subgroup).**
(DOCX)

## Author Contributions

**Conceptualization:** Hayman Win, Günther Fink.

**Data curation:** Sohana Shafique.

**Formal analysis:** Hayman Win, Günther Fink.

**Investigation:** Sohana Shafique.

**Methodology:** Hayman Win, Günther Fink.

**Visualization:** Hayman Win, Günther Fink.

**Writing – original draft:** Hayman Win.

**Writing – review & editing:** Hayman Win, Sohana Shafique, Nicole Probst-Hensch, Günther Fink.

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
