## [Decision Letter · Decision Letter 0]

23 Mar 2022

PGPH-D-22-00282

Change in nutritional status of urban slum children before and after the first COVID-19 wave in Bangladesh: a repeated cross-sectional assessment

Dear Dr. Win,

Thank you for submitting your manuscript to PLOS Global Public Health. After careful consideration, we feel that it has merit but does not fully meet PLOS Global Public Health’s publication criteria as it currently stands. Therefore, we invite you to submit a revised version of the manuscript that addresses the points raised during the review process.

We look forward to receiving your revised manuscript.

Kind regards,

Kalkidan Hassen Abate, PhD

Academic Editor

Journal Requirements:

2. In the online submission form, you indicated that your data is available only on request from a third party. Please note that your Data Availability Statement is currently missing [the name of the third party contact or institution / contact details for the third party, such as an email address or a link to where data requests can be made]. Please update your statement with the missing information. 

3. Please amend your detailed Financial Disclosure statement. This is published with the article, therefore should be completed in full sentences and contain the exact wording you wish to be published.

i) Please include all sources of funding (financial or material support) for your study. List the grants (with grant number) or organizations (with url) that supported your study, including funding received from your institution. 

ii). State the initials, alongside each funding source, of each author to receive each grant.

iii). State what role the funders took in the study. If the funders had no role in your study, please state: “The funders had no role in study design, data collection and analysis, decision to publish, or preparation of the manuscript.”

iv). If any authors received a salary from any of your funders, please state which authors and which funders.

Additional Editor Comments (if provided):

Dear Authors

The present article has important information to understand the impact of COVID-19 on child nutritional status. However, I have a major concern regarding the ethical issues related to the first survey undergone "before lockdown. Hence, a detail description of COVID protocol has to be provided. Information how subjects are protected from COVID during the survey has to be included.

Pleas address the comments of the reviewers as well.

Regards

Reviewers' comments:

Reviewer's Responses to Questions

**Comments to the Author**

1. Does this manuscript meet PLOS Global Public Health’s publication criteria? Is the manuscript technically sound, and do the data support the conclusions? The manuscript must describe methodologically and ethically rigorous research with conclusions that are appropriately drawn based on the data presented.

Reviewer #1: Yes

Reviewer #2: Yes

2. Has the statistical analysis been performed appropriately and rigorously?

Reviewer #1: Yes

Reviewer #2: Yes

3. Have the authors made all data underlying the findings in their manuscript fully available (please refer to the Data Availability Statement at the start of the manuscript PDF file)?

Reviewer #1: Yes

Reviewer #2: Yes

4. Is the manuscript presented in an intelligible fashion and written in standard English?

Reviewer #1: Yes

Reviewer #2: Yes

5. Review Comments to the Author

Reviewer #1: The MS is actual and new. It can be reccomended for publication. The results are corresponded to COVID-19 pandemic. The article demostrated how nutritiona status of children is changing due to pandemic.

Reviewer #2: Summary: Thank you for conducting the cross-sectional study related to the nutritional status of children in Bangladesh during the pandemic. The study results are useful for the government, healthcare providers, policymakers…etc. for formulating different strategies in their fields. I have the following suggestions for the manuscript.

The comments are listed below.

Specific comments:

Data collection and ethical approval

1. Provide the calculation and determination of the sample size.

2. Specify the questionnaires are self-designed or adopted from another study. Provide detailed information related to the questionnaire used in the current study. For example, if the questionnaires are self-designed, state the information related to the validity and reliability of the questionnaires, and any professional to validate the questions. If it is adopted from other studies, please state the questionnaires are free to use or approval have been obtained.

Limitations

3. Provide a sentence explaining the solution related to each limitation stated in this section.

Others

4. Suggested to provide a research checklist (e.g. STROBE statement for reporting a cross-sectional study) as supplementary information (OPTIONAL).

6. PLOS authors have the option to publish the peer review history of their article (what does this mean?). If published, this will include your full peer review and any attached files.

**Do you want your identity to be public for this peer review?** For information about this choice, including consent withdrawal, please see our Privacy Policy.

Reviewer #1: No

Reviewer #2: **Yes: **LEUNG Yuen Ling

---

## [Editor Report · Decision Letter 1]

20 May 2022

Change in nutritional status of urban slum children before and after the first COVID-19 wave in Bangladesh: a repeated cross-sectional assessment

PGPH-D-22-00282R1

Dear Ms. Win,

We are pleased to inform you that your manuscript 'Change in nutritional status of urban slum children before and after the first COVID-19 wave in Bangladesh: a repeated cross-sectional assessment' has been provisionally accepted for publication in PLOS Global Public Health.

Best regards,

Kalkidan Hassen Abate, PhD

Academic Editor